# Teaching Machines to Describe Images via Natural Language Feedback

**Huan Ling[1], Sanja Fidler[1,2]**
University of Toronto[1], Vector Institute[2]
{linghuan,fidler}@cs.toronto.edu

## Abstract

Robots will eventually be part of every household. It is thus critical to enable algorithms to learn from and be guided by non-expert users. In this paper, we bring a human in the loop, and enable a human teacher to give feedback to a learning agent in the form of natural language. We argue that a descriptive sentence can provide a much stronger learning signal than a numeric reward in that it can easily point to where the mistakes are and how to correct them. We focus on the problem of image captioning in which the quality of the output can easily be judged by non-experts. In particular, we first train a captioning model on a subset of images paired with human written captions. We then let the model describe new images and collect human feedback on the generated descriptions. We propose a hierarchical phrase-based captioning model, and design a *feedback network* that provides reward to the learner by conditioning on the human-provided feedback. We show that by exploiting descriptive feedback on new images our model learns to perform better than when given human written captions on these images.

## 1 Introduction

In the era where A.I. is slowly finding its way into everyone's lives, be in the form of social bots [36, 2], personal assistants [24, 13, 32], or household robots [1], it becomes critical to allow non-expert users to teach and guide their robots [37, 18]. For example, if a household robot keeps bringing food served on an ashtray thinking it's a plate, one should ideally be able to educate the robot about its mistakes, possibly without needing to dig into the underlying software.

Reinforcement learning has become a standard way of training artificial agents that interact with an environment. There have been significant advances in a variety of domains such as games [31, 25], robotics [17], and even fields like vision and NLP [30, 19]. RL agents optimize their action policies so as to maximize the expected reward received from the environment. Training typically requires a large number of episodes, particularly in environments with large action spaces and sparse rewards.

Several works explored the idea of incorporating humans in the learning process, in order to help the reinforcement learning agent to learn faster [35, 12, 11, 6, 5]. In most cases, a human teacher observes the agent act in an environment, and is allowed to give additional guidance to the learner. This feedback typically comes in the form of a simple numerical (or "good"/"bad") reward which is used to either shape the MDP reward [35] or directly shape the policy of the learner [5].

In this paper, we aim to exploit natural language as a way to guide an RL agent. We argue that a sentence provides a much stronger learning signal than a numeric reward in that it can easily point to where the mistakes occur and suggests how to correct them. Such descriptive feedback can thus naturally facilitate solving the credit assignment problem as well as to help guide exploration. Despite its clear benefits, very few approaches aimed at incorporating language in Reinforcement Learning. In pioneering work, [22] translated natural language advice into a short program which was used to bias action selection. While this is possible in limited domains such as in navigating a maze [22] or learning to play a soccer game [15], it can hardly scale to the real scenarios with large action spaces requiring versatile language feedback.

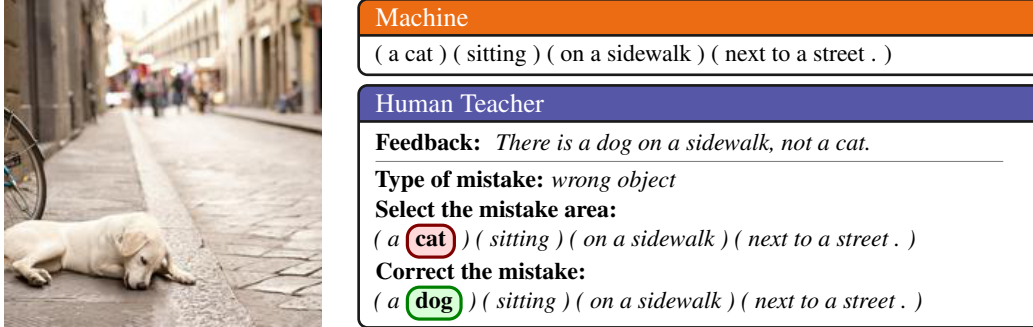

Figure 1: Our model accepts feedback from a human teacher in the form of natural language. We generate captions using the current snapshot of the model and collect feedback via AMT. The annotators are requested to focus their feedback on a single word/phrase at a time. Phrases, indicated with brackets in the example, are part or our captioning model's output. We also collect information about which word the feedback applies to and its suggested correction. This information is used to train our *feedback network*.

Here our goal is to allow a non-expert human teacher to give feedback to an RL agent in the form of natural language, just as one would to a learning child. We focus on the problem of image captioning in which the quality of the output can easily be judged by non-experts.

Towards this goal, we make several contributions. We propose a hierarchical phrase-based RNN as our image captioning model, as it can be naturally integrated with human feedback. We design a web interface which allows us to collect natural language feedback from human "teachers" for a snapshot of our model, as in Fig. 1. We show how to incorporate this information in Policy Gradient RL [30], and show that we can improve over RL that has access to the same amount of ground-truth captions. Our code and data will be released (http://www.cs.toronto.edu/~linghuan/feedbackImageCaption/) to facilitate more human-like training of captioning models.

## 2   Related Work

Several works incorporate human feedback to help an RL agent learn faster. [35] exploits humans in the loop to teach an agent to cook in a virtual kitchen. The users watch the agent learn and may intervene at any time to give a scalar reward. Reward shaping [26] is used to incorporate this information in the MDP. [6] iterates between "practice", during which the agent interacts with the real environment, and a critique session where a human labels any subset of the chosen actions as good or bad. In [12], the authors compare different ways of incorporating human feedback, including reward shaping, Q augmentation, action biasing, and control sharing. The same authors implement their TAMER framework on a real robotic platform [11]. [5] proposes policy shaping which incorporates right/wrong feedback by utilizing it as direct policy labels. These approaches mostly assume that humans provide a numeric reward, unlike in our work where the feedback is given in natural language.

A few attempts have been made to advise an RL agent using language. [22]'s pioneering work translated advice to a short program which was then implemented as a neural network. The units in this network represent Boolean concepts, which recognize whether the observed state satisfies the constraints given by the program. In such a case, the advice network will encourage the policy to take the suggested action. [15] incorporated natural language advice for a RoboCup simulated soccer task. They too translate the advice in a formal language which is then used to bias action selection. Parallel to our work, [7] exploits textual advice to improve training time of the A3C algorithm in playing an Atari game. Recently, [37, 18] incorporates human feedback to improve a text-based QA agent. Our work shares similar ideas, but applies them to the problem of image captioning. In [27], the authors incorporate human feedback in an active learning scenario, however not in an RL setting.

Captioning represents a natural way of showing that our algorithm understands a photograph to a non-expert observer. This domain has received significant attention [8, 39, 10], achieving impressive performance on standard benchmarks. Our phrase model shares the most similarity with [16], but differs in that exploits attention [39], linguistic information, and RL to train. Several recent approaches trained the captioning model with policy gradients in order to directly optimize for the desired performance metrics [21, 30, 3]. We follow this line of work. However, to the best of our knowledge, our work is the first to incorporate natural language feedback into a captioning model.

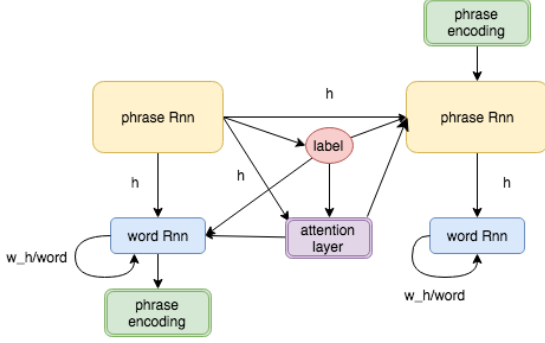

Figure 2: Our hierarchical phrase-based captioning model, composed of a phrase-RNN at the top level, and a word-level RNN which outputs a sequence of words for each phrase. The useful property of this model is that it directly produces an output sentence segmented into linguistic phrases. We exploit this information while collecting and incorporating human feedback into the model. Our model also exploits attention, and linguistic information (phrase labels such as noun, preposition, verb, and conjunction phrase). Please see text for details.

Related to our efforts is also work on dialogue based visual representation learning [40, 41], however this work tackles a simpler scenario, and employs a slightly more engineered approach.

We stress that our work differs from the recent efforts in conversation modeling [19] or visual dialog [4] using Reinforcement Learning. These models aim to mimic human-to-human conversations while in our work the human converses with and guides an artificial learning agent.

# 3 Our Approach

Our framework consists of a new phrase-based captioning model trained with Policy Gradients that incorporates natural language feedback provided by a human teacher. While a number of captioning methods exist, we design our own which is phrase-based, allowing for natural guidance by a non-expert. In particular, we argue that the strongest learning signal is provided when the feedback describes one mistake at a time, e.g. a single wrong word or a phrase in a caption. An example can be seen in Fig. 1. This is also how one most effectively teaches a learning child. To avoid parsing the generated sentences at test time, we aim to predict phrases directly with our captioning model. We first describe our phrase-based captioner, then describe our feedback collection process, and finally propose how to exploit feedback as a guiding signal in policy gradient optimization.

## 3.1 Phrase-based Image Captioning

Our captioning model, forming the base of our approach, uses a hierarchical Recurrent Neural Network, similar to [34, 14]. In [14], the authors use a two-level LSTM to generate paragraphs, while [34] uses it to generate sentences as a sequence of phrases. The latter model shares a similar overall structure as ours, however, our model additionally reasons about the type of phrases and exploits the attention mechanism over the image.

The structure of our model is best explained through Fig. 2. The model receives an image as input and outputs a caption. It is composed of a *phrase RNN* at the top level, and a *word RNN* that generates a sequence of words for each phrase. One can think of the phrase RNN as providing a "topic" at each time step, which instructs the word RNN what to talk about.

Following [39], we use a convolutional neural network in order to extract a set of feature vectors $a = (\mathbf{a}_1, \ldots, \mathbf{a}_n)$, with $\mathbf{a}_j$ a feature in location $j$ in the input image. We denote the hidden state of the phrase RNN at time step $t$ with $h_t$, and $h_{t,i}$ to denote the $i$-th hidden state of the word RNN for the $t$-th phrase. Computation in our model can be expressed with the following equations:

$$
\text{phrase-RNN} \begin{cases}
h_t = f_{phrase}(h_{t-1}, l_{t-1}, c_{t-1}, e_{t-1}) \\
l_t = \text{softmax}(f_{phrase-label}(h_t)) \\
c_t = f_{att}(h_t, l_t, a) \\
h_{t,0} = f_{phrase-word}(h_t, l_t, c_t)
\end{cases}
$$

$$
\text{word-RNN} \begin{cases}
h_{t,i} = f_{word}(h_{t,i-1}, c_t, w_{t,i}) \\
w_{t,i} = f_{out}(h_{t,i}, c_t, w_{t,i-1}) \\
e_t = f_{word-phrase}(w_{t,1}, \ldots, w_{t,end})
\end{cases}
$$

| | |
|---|---|
| $f_{phrase}$ | LSTM, dim 256 |
| $f_{phrase-label}$ | 3-layer MLP |
| $f_{att}$ | 2-layer MLP with ReLu |
| $f_{phrase-word}$ | 3-layer MLP with ReLu |
| $f_{word}$ | LSTM, dim 256 |
| $f_{out}$ | deep decoder [28] |
| $f_{word-phrase}$ | mean+3-lay. MLP with ReLu |

| Image | Ref. caption | Feedback | Corr. caption | Image | Ref. caption | Feedback | Corr. caption |
|---|---|---|---|---|---|---|---|
| 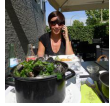 | ( a woman ) ( is sitting ) ( on a bench ) ( with a plate ) ( of food . ) | What the woman is sitting on is not visible. | ( a woman ) ( is sitting ) ( with a plate ) ( of food . ) |  | ( a man ) ( riding a motorcycle ) ( on a city street . ) | There is a man and a woman. | ( a man and a woman ) ( riding a motorcycle ) ( on a city street . ) |
|  | ( a horse ) ( is standing ) ( in a barn ) ( in a field . ) | There is no barn. There is a fence. | ( a horse ) ( is standing ) ( in a fence ) ( in a field . ) |  | ( a man ) ( is swinging a baseball bat ) ( on a field . ) | The baseball player is not swinging a bate. | ( a man ) ( is playing baseball ) ( on a field . ) |

Table 1: Examples of collected feedback. Reference caption comes from the MLE model.

Table 2: Statistics for our collected feedback information. The table on the right shows how many times the feedback sentences mention words to be corrected and suggest correction.

| | | | |
|---|---|---|---|
| Num. of evaluated examples (annot. round 1) | 9000 | Something should be replaced | 2999 |
| Evaluated as containing errors | 5150 | mistake word is in description | 2664 |
| To ask for feedback (annot. round 2) | 4174 | correct word is in description | 2674 |
| Avg. num. of feedback rounds per image | 2.22 | Something is missing | 334 |
| Avg. num. of words in feedback sent. | 8.04 | missing word is in description | 246 |
| Avg. num. of words needing correction | 1.52 | Something should be removed | 841 |
| Avg. num. of modified words | 1.46 | removed word is in description | 779 |

feedback round: number of correction rounds for the same example, description: natural language feedback

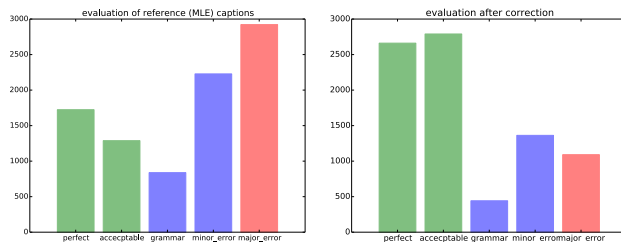

Figure 3: Caption quality evaluation by the human annotators. Plot on the left shows evaluation for captions generated with our reference model (MLE). The right plot shows evaluation of the human-corrected captions (after completing at least one round of feedback).

As in [39], $c_t$ denotes a context vector obtained by applying the attention mechanism to the image. This context vector essentially represents the image area that the model "looks at" in order to generate the $t$-th phrase. This information is passed to both the word-RNN as well as to the next hidden state of the phrase-RNN. We found that computing two different context vectors, one passed to the phrase and one to the word RNN, improves generation by 0.6 points (in weighted metric, see Table 4) mainly helping the model to avoid repetition of words. Furthermore, we noticed that the quality of attention significantly improves (1.5 points, Table 4) if we provide it with additional linguistic information. In particular, at each time step $t$ our phrase RNN also predicts a phrase label $l_t$, following the standard definition from the Penn Tree Bank. For each phrase, we predict one out of four possible phrase labels, i.e., a noun (NP), preposition (PP), verb (VP), and a conjunction phrase (CP). We use additional <EOS> token to indicate the end of the sentence. By conditioning on the NP label, we help the model look at the objects in the image, while VP may focus on more global image information.

Above, $w_{t,i}$ denotes the $i$-th word output of the word-RNN in the $t$-th phrase, encoded with a one-hot vector. Note that we use an additional <EOP> token in word-RNN's vocabulary, which signals the end-of-phrase. Further, $e_t$ encodes the generated phrase via simple mean-pooling over the words, which provides additional word-level context to the next phrase. Details about the choices of the functions are given in the table. Following [39], we use a deep output layer [28] in the LSTM and double stochastic attention.

**Implementation details.** To train our hierarchical model, we first process MS-COCO image caption data [20] using the Stanford Core NLP toolkit [23]. We flatten each parse tree, separate a sentence into parts, and label each part with a phrase label (<NP>, <PP>, <CP>, <VP>). To simplify the phrase structure, we merge some NPs to its previous phrase label if it is not another NP.

**Pre-training.** We pre-train our model using the standard cross-entropy loss. We use the ADAM optimizer [9] with learning rate 0.001. We discuss Policy Gradient optimization in Subsec. 3.4.

### 3.2 Crowd-sourcing Human Feedback

We aim to bring a human in the loop when training the captioning model. Towards this, we create a web interface that allows us to collect feedback information on a larger scale via AMT. Our interface

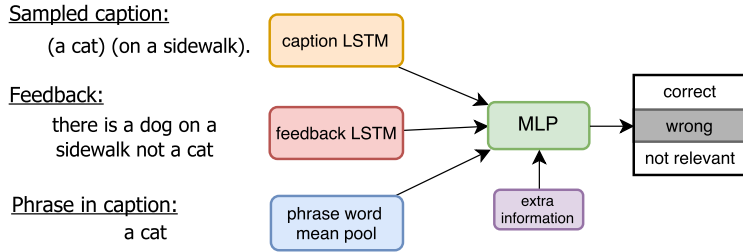

**Sampled caption:**
(a cat) (on a sidewalk).

**Feedback:**
there is a dog on a sidewalk not a cat

**Phrase in caption:**
a cat

Figure 4: The architecture of our feedback network (FBN) that classifies each phrase (bottom left) in a sampled sentence (top left) as either *correct*, *wrong* or *not relevant*, by conditioning on the feedback sentence.

is akin to that depicted in Fig. 1, and we provide further visualizations in the Appendix. We also provide it online on our project page. In particular, we take a snapshot of our model and generate captions for a subset of MS-COCO images [20] using greedy decoding. In our experiments, we take the model trained with the MLE objective.

We do two rounds of annotation. In the first round, the annotator is shown a captioned image and is asked to assess the quality of the caption, by choosing between: *perfect*, *acceptable*, *grammar mistakes only*, *minor* or *major* errors. We asked the annotators to choose minor and major error if the caption contained errors in semantics, i.e., indicating that the "robot" is not understanding the photo correctly. We advised them to choose *minor* for small errors such as wrong or missing attributes or awkard prepositions, and go with major errors whenever any object or action naming is wrong.

For the next (more detailed, and thus more costly) round of annotation, we only select captions which are not marked as either perfect or acceptable in the first round. Since these captions contain errors, the new annotator is required to provide detailed feedback about the mistakes. We found that some of the annotators did not find errors in some of these captions, pointing to the annotator noise in the process. The annotator is shown the generated caption, delineating different phrases with the "(" and ")" tokens. We ask the annotator to 1) choose the type of required correction, 2) write feedback in natural language, 3) mark the type of mistake, 4) highlight the word/phrase that contains the mistake, 5) correct the chosen word/phrase, 6) evaluate the quality of the caption after correction. We allow the annotator to submit the HIT after one correction even if her/his evaluation still points to errors. However, we plea to the good will of the annotators to continue in providing feedback. In the latter case, we reset the webpage, and replace the generated caption with their current correction.

The annotator first chooses the type of error, i.e., something " should be replaced", "is missing", or "should be deleted". (S)he then writes a sentence providing feedback about the mistake and how it should be corrected. We require that the feedback is provided sequentially, describing a single mistake at a time. We do this by restricting the annotator to only select mistaken words within a single phrase (in step 4). In 3), the annotator marks further details about the mistake, indicating whether it corresponds to an error in *object*, *action*, *attribute*, *preposition*, *counting*, or *grammar*. For 4) and 5) we let the annotator highlight the area of mistake in the caption, and replace it with a correction.

The statistics of the data is provided in Table 2, with examples shown in Table 1. An interesting fact is that the feedback sentences in most cases mention both the wrong word from the caption, as well as the correction word. Fig. 3 (left) shows evaluation of the caption quality of the reference (MLE) model. Out of 9000 captions, 5150 are marked as containing errors (either semantic or grammar), and we randomly choose 4174 for the second round of annotation (detailed feedback). Fig. 3 (left) shows the quality of all the captions after correction, i.e. good reference captions as well as 4174 corrected captions as submitted by the annotators. Note that we only paid for one round of feedback, thus some of the captions still contained errors even after correction. Interestingly, on average the annotators still did 2.2 rounds of feedback per image (Table 2).

### 3.3 Feedback Network

Our goal is to incorporate natural language feedback into the learning process. The collected feedback contains rich information of how the caption can be improved: it conveys the location of the mistake and typically suggests how to correct it, as seen in Table 2. This provides strong supervisory signal which we want to exploit in our RL framework. In particular, we design a neural network which will provide additional reward based on the feedback sentence. We refer to it as the *feedback network* (FBN). We first explain our feedback network, and show how to integrate its output in RL.

| Sampled caption | Feedback | Phrase | Prediction |
|---|---|---|---|
| A cat on a sidewalk. | | A cat | wrong |
| A dog on a sidewalk. | There is a dog on a sidewalk not a cat. | A dog | correct |
| A cat on a sidewalk. | | on a sidewalk | not relevant |

Table 3: Example classif. of each phrase in a newly sampled caption into correct/wrong/not-relevant conditioned on the feedback sentence. Notice that we do not need the image to judge the correctness/relevance of a phrase.

Note that RL training will require us to generate samples (captions) from the model. Thus, during training, the sampled captions for each training image will change (will differ from the reference MLE caption for which we obtained feedback for). The goal of the feedback network is to read a newly sampled caption, and judge the correctness of each phrase conditioned on the feedback. We make our FBN to only depend on text (and not on the image), making its learning task easier. In particular, our FBN performs the following computation:

$$h_t^{caption} = f_{sent}(h_{t-1}^{caption}, w_t^c) \qquad (1)$$

$$h_t^{feedback} = f_{sent}(h_{t-1}^{feedback}, w_t^f) \qquad (2)$$

$$q_i = f_{phrase}(w_{i,1}^c, \ldots, w_{i,N}^c) \qquad (3)$$

$$o_i = f_{fbn}(h_T^c, h_{T'}^f, q_i, m) \qquad (4)$$

| $f_{sent}$ | LSTM, dim 256 |
|---|---|
| $f_{phrase}$ | linear+mean pool |
| $f_{fbn}$ | 3-layer MLP with dropout +3-way softmax |

Here, $w_t^c$ and $w_t^f$ denote the one-hot encoding of words in the sampled caption and feedback sentence, respectively. By $w_{i,.}^c$ we denote words in the $i$-th phrase of the sampled caption. FBN thus encodes both the caption and feedback using an LSTM (with shared parameters), performs mean pooling over the words in a phrase to represent the phrase $i$, and passes this information through a 3-layer MLP. The MLP additionally accepts information about the mistake type (e.g., wrong object/action) encoded as a one hot vector $m$ (denoted as "extra information" in Fig. 4). The output layer of the MLP is a 3-way classification layer that predicts whether the phrase $i$ is *correct*, *wrong*, or *not relevant* (wrt feedback sentence). An example output from FBN is shown in Table 3.

**Implementation details.** We train our FBN with the ground-truth data that we collected. In particular, we use (reference, feedback, marked phrase in reference caption) as an example of a *wrong* phrase, (corrected sentence, feedback, marked phrase in corrected caption) as an example of the *correct* phrase, and treat the rest as the *not relevant* label. Reference here means the generated caption that we collected feedback for, and marked phrase means the phrase that the annotator highlighted in either the reference or the corrected caption. We use the standard cross-entropy loss to train our model. We use ADAM [9] with learning rate 0.001, and a batch size of 256. When a reference caption has several feedback sentences, we treat each one as independent training data.

## 3.4 Policy Gradient Optimization using Natural Language Feedback

We follow [30, 29] to directly optimize for the desired image captioning metrics using the Policy Gradient technique. For completeness, we briefly summarize it here [30].

One can think of an caption decoder as an agent following a parameterized policy $p_\theta$ that selects an action at each time step. An "action" in our case requires choosing a word from the vocabulary (for the word RNN), or a phrase label (for the phrase RNN). An "agent" (our captioning model) then receives the reward after generating the full caption, i.e., the reward can be any of the automatic metrics, their weighted sum [30, 21], or in our case will also include the reward from feedback.

The objective for learning the parameters of the model is the expected reward received when completing the caption $w^s = (w_1^s, \ldots, w_T^s)$ ($w_t^s$ is the word sampled from the model at time step $t$):

$$L(\theta) = -E_{w^s \sim p_\theta}[r(w^s)] \qquad (5)$$

To optimize this objective, we follow the reinforce algorithm [38], as also used in [30, 29]. The gradient of (5) can be computed as

$$\nabla_\theta L(\theta) = -E_{w^s \sim p_\theta}[r(w^s)\nabla_\theta \log p_\theta(w^s)], \qquad (6)$$

which is typically estimated by using a single Monte-Carlo sample:

$$\nabla_\theta L(\theta) \approx -r(w^s)\nabla_\theta \log p_\theta(w^s) \qquad (7)$$

We follow [30] to define the baseline $b$ as the reward obtained by performing greedy decoding:

$$b = r(\hat{w}), \quad \hat{w}_t = \arg\max p(w_t|h_t)$$
$$\nabla_\theta L(\theta) \approx -(r(w^s) - r(\hat{w}))\nabla_\theta \log p_\theta(w^s) \tag{8}$$

Note that the baseline does not change the expected gradient but can drastically reduce its variance.

**Reward.** We define two different rewards, one at the sentence level (optimizing for a performance metrics), and one at the phrase level. We use human feedback information in both. We first define the sentence reward wrt to a reference caption as a weighted sum of the BLEU scores:

$$r(w^s) = \beta \sum_i \lambda_i \cdot BLEU_i(w^s, ref) \tag{9}$$

In particular, we choose $\lambda_1 = \lambda_2 = 0.5$, $\lambda_3 = \lambda_4 = 1$, $\lambda_5 = 0.3$. As reference captions to compute the reward, we either use the reference captions generated by a snapshot of our model which were evaluated as not having minor and major errors, or ground-truth captions. The details are given in the experimental section. We weigh the reward by the caption quality as provided by the annotators. In particular, $\beta = 1$ for *perfect* (or GT), $0.8$ for *acceptable*, and $0.6$ for *grammar/fluency issues only*.

We further incorporate the reward provided by the feedback network. In particular, our FBN allows us to define the reward at the phrase level (thus helping with the credit assignment problem). Since our generated sentence is segmented into phrases, i.e., $w^s = w_1^p w_2^p \dots w_P^p$, where $w_t^p$ denotes the (sequence of words in the) $t$-th phrase, we define the combined phrase reward as:

$$r(w_t^p) = r(w^s) + \lambda_f f_{fbn}(w^s, feedback, w_t^p) \tag{10}$$

Note that FBN produces a classification of each phrase. We convert this into reward, by assigning *correct* to 1, *wrong* to $-1$, and 0 to *not relevant*. We do not weigh the reward by the confidence of the network, which might be worth exploring in the future. Our final gradient takes the following form:

$$\nabla_\theta L(\theta) = -\sum_{p=1}^{P} (r(w^p) - r(\hat{w}^p))\nabla_\theta \log p_\theta(w^p) \tag{11}$$

**Implementation details.** We use Adam with learning rate $1e^{-6}$ and batch size 50. As in [29], we follow an annealing schedule. We first optimize the cross entropy loss for the first $K$ epochs, then for the following $t = 1, \dots, T$ epochs, we use cross entropy loss for the first $(P - floor(t/m))$ phrases (where $P$ denotes the number of phrases), and the policy gradient algorithm for the remaining $floor(t/m)$ phrases. We choose $m = 5$. When a caption has multiple feedback sentences, we take the sum of the FBN's outputs (converted to rewards) as the reward for each phrase. When a sentence does not have any feedback, we assign it a zero reward.

## 4 Experimental Results

To validate our approach we use the MS-COCO dataset [20]. We use 82K images for training, 2K for validation, and 4K for testing. In particular, we randomly chose 2K val and 4K test images from the official validation split. To collect feedback, we randomly chose 7K images from the training set, as well as all 2K images from our validation. In all experiments, we report the performance on our (held out) test set. For all the models (including baselines) we used a pre-trained VGG [33] network to extract image features. We use a word vocabulary size of 23,115.

**Phrase-based captioning model.** We analyze different instantiations of our phrase-based captioning in Table 4, showing the importance of predicting phrase labels. To sanity check our model we compare it to a flat approach (word-RNN only) [39]. Overall, our model performs slightly worse than [39] (0.66 points). However, the main strength of our model is that it allows a more natural integration with feedback. Note that these results are reported for the models trained with MLE.

**Feedback network.** As reported in Table 2, our dataset which contains detailed feedback (descriptions) contains 4173 images. We randomly select 9/10 of them to serve as a training set for our feedback network, and use 1/10 of them to be our test set. The classification performance of our FBN is reported in Table 5. We tried exploiting additional information in the network. The second line reports the result for FBN which also exploits the reference caption (for which the feedback was written) as input, represented with a LSTM. The model in the third line uses the type of error, i.e. the phrase is "missing", "wrong", or "redundant". We found that by using information about what kind of mistake the reference caption had (e.g., corresponding to misnaming an *object*, *action*, etc) achieves the best performance. We use this model as our FBN used in the following experiments.

|                                  | BLEU-1 | BLEU-2 | BLEU-3 | BLEU-4 | ROUGE-L | Weighted metric |
|----------------------------------|--------|--------|--------|--------|---------|-----------------|
| flat (word level) with att       | 65.36  | 44.03  | 29.68  | 20.40  | 51.04   | **104.78**      |
| phrase with att.                 | 64.69  | 43.37  | 28.80  | 19.31  | 50.80   | 102.14          |
| phrase with att +phrase label    | 65.46  | 44.59  | 29.36  | 19.25  | 51.40   | 103.64          |
| phrase with 2 att +phrase label  | 65.37  | 44.02  | 29.51  | 19.91  | 50.90   | 104.12          |

Table 4: Comparing performance of the flat captioning model [39], and different instantiations of our phrase-based captioning model. All these models were trained using the cross-entropy loss.

| Feedback network                        | Accuracy |
|-----------------------------------------|----------|
| no extra information                    | 73.30    |
| use reference caption                   | 73.24    |
| use "missing"/"wrong"/"redundant"       | 72.92    |
| use "action"/"object"/"preposition"/etc | **74.66**|

Table 5: Classification results of our feedback network (FBN) on a held-out feedback data. The FBN predicts *correct*/*wrong*/*not relevant* for each phrase in a caption. See text for details.

**RL with Natural Language Feedback.** In Table 6 we report the performance for several instantiations of the RL models. All models have been pre-trained using cross-entropy loss (MLE) on the full MS-COCO training set. For the next rounds of training, all the models are trained only on the 9K images that comprise our full evaluation+feedback dataset from Table 2. In particular, we separate two cases. In the first, standard case, the "agent" has access to 5 captions for each image. We experiment with different types of captions, e.g. ground-truth captions (provided by MS-COCO), as well as feedback data. For a fair comparison, we ensure that each model has access to (roughly) the same amount of data. This means that we count a feedback sentence as one source of information, and a human-corrected reference caption as yet another source. We also exploit reference (MLE) captions which were evaluated as correct, as well as corrected captions obtained from the annotators. In particular, we tried two types of experiments. We define "C" captions as all captions that were corrected by the annotators and were not evaluated as containing *minor* or *major* error, and ground-truth captions for the rest of the images. For "A", we use all captions (including reference MLE captions) that did not have *minor* or *major* errors, and GT for the rest. A detailed break-down of these captions is reported in Table 7.

We first test a model using the standard cross-entropy loss, but which now also has access to the corrected captions in addition to the 5GT captions. This model (MLEC) is able to improve over the original MLE model by $1.4$ points. We then test the RL model by optimizing the metric wrt the 5GT captions (as in [30]). This brings an additional point, achieving $2.4$ over the MLE model. Our RL agent with feedback is given access to 3GT captions, the "C" captions and feedback sentences. We show that this model outperforms the no-feedback baseline by $0.5$ points. Interestingly, with "A" captions we get an additional $0.3$ boost. If our RL agent has access to 4GT captions and feedback descriptions, we achieve a total of $1.1$ points over the baseline RL model and $3.5$ over the MLE model. Examples of generated captions are shown in Fig. 6.

We also conducted a human evaluation using AMT. In particular, Turkers are shown an image captioned by the baseline RL and our method, and are asked to choose the better caption. As shown in Fig. 5, our RL with feedback is $4.7$ percent higher than the RL baseline. We additionally count how much human interaction is required for either the baseline RL and our approach. In particular, we count every interaction with the keyboard as 1 click. In evaluation, choosing the quality of the caption counts as 1 click, and for captions/feedback, every letter counts as a click. The main save comes from the first evaluation round, in which we only as for the quality of captions. Overall, there is almost half clicks saved in our setting.

We also test a more realistic scenario, in which the models have access to either a single GT caption, or in our case "C" (or "A") and feedback. This mimics a scenario in which the human teacher observes the agent and either gives feedback about the agent's mistakes, or, if the agent's caption is completely wrong, the teacher writes a new caption. Interestingly, RL when provided with the corrected captions performs better than when given GT captions. Overall, our model outperforms the base RL (no feedback) by $1.2$ points. We note that our RL agents are trained (not counting pre-training) only on a small (9K) subset of the full MS-COCO training set. Further improvements are thus possible.

**Discussion.** These experiments make an important point. Instead of giving the RL agent a completely new target (caption), a better strategy is to "teach" the agent about the mistakes it is doing and suggest a correction. Natural language thus offers itself as a rich modality for providing such guidance not only to humans but also to artificial agents.

Table 6: Comparison of our RL with feedback information to baseline RL and MLE models.

| | | BLEU-1 | BLEU-2 | BLEU-3 | BLEU-4 | ROUGE-L | Weighted metric |
|---|---|---|---|---|---|---|---|
| 5 sent. | MLE (5 GT) | 65.37 | 44.02 | 29.51 | 19.91 | 50.90 | 104.12 |
| | MLEC (5 GT + C) | 66.85 | 45.19 | 29.89 | 19.79 | 51.20 | 105.58 |
| | MLEC (5 GT + A) | 66.14 | 44.87 | 30.17 | 20.27 | 51.32 | 105.47 |
| | RLB (5 GT) | 66.90 | 45.10 | 30.10 | 20.30 | 51.10 | 106.55 |
| | RLF (3GT+FB+C) | 66.52 | 45.23 | 30.48 | **20.66** | 51.41 | 107.02 |
| | RLF (3GT+FB+A) | 66.98 | 45.54 | 30.52 | 20.53 | **51.54** | 107.31 |
| | RLF (4GT + FB) | **67.10** | **45.50** | **30.60** | 20.30 | 51.30 | **107.67** |
| 1 sent. | RLB (1 GT) | 65.68 | 44.58 | 29.81 | 19.97 | 51.07 | 104.93 |
| | RLB (C) | 65.84 | 44.64 | 30.01 | 20.23 | 51.06 | 105.50 |
| | RLB (A) | 65.81 | 44.58 | 29.87 | 20.24 | 51.28 | 105.31 |
| | RLF (C + FB) | 65.76 | 44.65 | **30.20** | **20.62** | 51.35 | 106.03 |
| | RLF (A + FB) | **66.23** | **45.00** | 30.15 | 20.34 | **51.58** | **106.12** |

**GT**: ground truth captions; **FB**: feedback; **MLE(A)(C)**: MLE model using five GT sentences + either C or A captions (see text and Table 7); **RLB**: baseline RL (no feedback network); **RLF**: RL with feedback (here we also use C or A captions as well as FBN);

| | ground-truth | perfect | acceptable | grammar error only |
|---|---|---|---|---|
| A | 3107 | 2661 | 2790 | 442 |
| C | 6326 | 1502 | 1502 | 234 |

Table 7: Detailed break-down of what captions were used as "A" or "C" in Table 6 for computing additional rewards in RL.

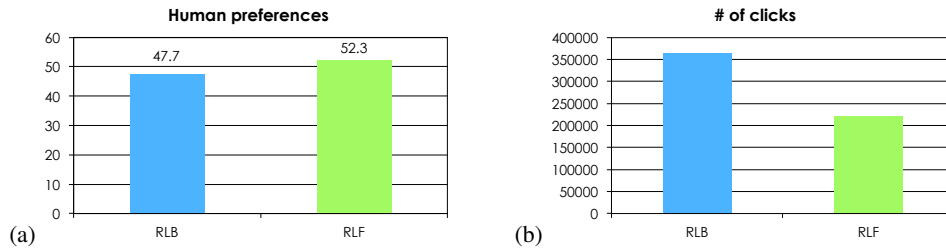

Figure 5: (a) Human preferences: RL baseline vs RL with feedback (our approach), (b) Number of human "clicks" required for MLE/baseline RL, and ours. A click is counted when an annotator hits the keyboard: in evaluation, choosing the quality of the caption counts as 1 click, and for captions/feedback, every letter counts as a click. The main save comes from the first evaluation round, in which we only as for the quality of captions.

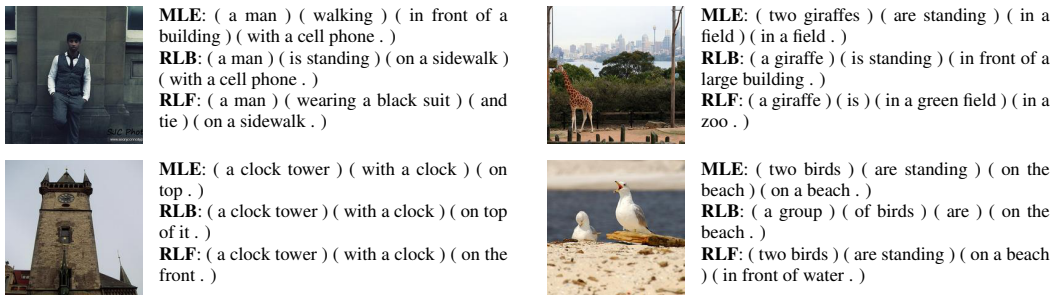

**MLE**: ( a man ) ( walking ) ( in front of a building ) ( with a cell phone . )
**RLB**: ( a man ) ( is standing ) ( on a sidewalk ) ( with a cell phone . )
**RLF**: ( a man ) ( wearing a black suit ) ( and tie ) ( on a sidewalk . )

**MLE**: ( two giraffes ) ( are standing ) ( in a field ) ( in a field . )
**RLB**: ( a giraffe ) ( is standing ) ( in front of a large building . )
**RLF**: ( a giraffe ) ( is ) ( in a green field ) ( in a zoo . )

**MLE**: ( a clock tower ) ( with a clock ) ( on top . )
**RLB**: ( a clock tower ) ( with a clock ) ( on top of it . )
**RLF**: ( a clock tower ) ( with a clock ) ( on the front . )

**MLE**: ( two birds ) ( are standing ) ( on the beach ) ( on a beach . )
**RLB**: ( a group ) ( of birds ) ( are ) ( on the beach . )
**RLF**: ( two birds ) ( are standing ) ( on a beach ) ( in front of water . )

Figure 6: Qualitative examples of captions from the MLE and RLB models (baselines), and our RBF model.

# 5  Conclusion

In this paper, we enable a human teacher to provide feedback to the learning agent in the form of natural language. We focused on the problem of image captioning. We proposed a hierarchical phrase-based RNN as our captioning model, which allowed natural integration with human feedback. We crowd-sourced feedback for a snapshot of our model, and showed how to incorporate it in Policy Gradient optimization. We showed that by exploiting descriptive feedback our model learns to perform better than when given independently written captions.

# Acknowledgment

We gratefully acknowledge the support from NVIDIA for their donation of the GPUs used for this research. This work was partially supported by NSERC. We also thank Relu Patrascu for infrastructure support.

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
