[Supplementary Material]

# Supplementary Material:
# Teaching Machines to Describe Images with Natural Language Feedback

**David S. Hippocampus**[*]
Department of Computer Science
Cranberry-Lemon University
Pittsburgh, PA 15213
`hippo@cs.cranberry-lemon.edu`

## 1   Feedback Crowd-Sourcing Interface

Figure 1: Our web-based feedback collection interface.

---

[*]Use footnote for providing further information about author (webpage, alternative address)—*not* for acknowledging funding agencies.

 ## 2 Examples of Collected Feedback

| Image | Before Correction | Feedback | After Correction |
|---|---|---|---|
|  | ( a man ) ( holding a hot dog ) ( in a restaurant . ) | The man is wearing a red sweater and this should be mentioned. | ( a man wearing a red sweater ) ( holding a hot dog ) ( in a restaurant . ) |
|  | ( a living room ) ( with a bed ) ( and a television . ) | There is no bed, only a couch. | (a living room ) ( with a couch ) ( and a television . ) |
|  | ( a man ) ( sitting ) ( on a bench ) ( with a cat ) ( on the bench . ) | there is a cup, but no cat on the bench. | ( a man ) ( sitting ) ( on a bench ) ( with a cup ) ( on the bench . ) |
|  | ( a bird ) ( is flying ) ( in the air ) ( with a frisbee . ) | There is no frisbee in the picture | ( a bird ) ( is flying ) ( in the air .) |

Table 1: Examples of Collected Feedback

## 3 Qualitative Examples

Figure 2: Examples of our phrase-based attention and phrase-label prediction.

| Image | Caption |
|---|---|
|  | • MLE: ( a computer ) ( sitting ) ( on top of a desk ) ( on a monitor . )<br>• RLB: ( a laptop ) ( sitting ) ( on top of a desk ) ( next to a computer monitor . )<br>• RLF: ( a computer ) ( sitting ) ( on top of a desk ) ( with a monitor . ) |
|  | • MLE: ( a suitcase ) ( is ) ( on a bed ) ( with a bag ) ( on top of it . )<br>• RLB: ( a suitcase ) ( is ) ( on a table ) ( with a suitcase . )<br>• RLF: ( a luggage bag ) ( sitting ) ( on a floor ) ( in a room . ) |

Table 2: Qualitative captioning results for our model and the baselines.

| Image | Caption |
|---|---|
|  | <ul><li>MLE: ( a red bus ) ( driving down a street ) ( with a person ) ( waiting ) ( on the street . )</li><li>RLB:( a red bus ) ( driving down a street ) ( with people ) ( driving ) ( on the side . )</li><li>RLF:( a red bus ) ( is driving down the street . )</li></ul> |
|  | <ul><li>MLE:( a street ) ( with a traffic light ) ( and a bus ) ( in the background . )</li><li>RLB:( a person ) ( walking ) ( on a city street ) ( with a yellow sign . )</li><li>RLF: ( a street ) ( with a car ) ( is driving down a street . )</li></ul> |

Table 3: caption results

| Image | Caption |
|---|---|
|  | <ul><li>MLE: ( a man ) ( is jumping ) ( into the air to ) ( catch a frisbee . )</li><li>RLB: ( a man ) ( is throwing a frisbee ) ( in a park . )</li><li>RLF: ( a man ) ( is holding a frisbee ) ( in his mouth . )</li></ul> |

Table 4: An example of a failed caption, where however the baselines produce a more reasonable caption.

4 **References**