[Reviews · NeurIPS 2017]

Reviewer 1



The paper presents a learning framework of incorporating human feedback in RL and applies the method to image captioning. specifically, the authors collect human feedback on machine’s caption output, including scoring (from perfect to major error) and correction (if not perfect or acceptable). Then authors train a feedback network to simulate human’s feedback assignment on the correctness of a phrase. Later, the authors use reinforcement learning to train the captioning model, with rewards computed from both auto-metrics (like weighted BLEU) and simulated human feedback based on the feedback network. The authors conducted experiments on COCO to evaluate the effectiveness of the model. Though a interesting task, the paper has room to improve, e.g., 1. The novelty of the paper is not very clear. The setting of including human in the loop for RL training is interesting. However, the actual implementation is not an online human-in-the-loop setting, instead it is a two-stage batch mode of human labeling. Therefore, the authors need to design a simulator (e.g., the feedback network) to estimate human feedback in RL training, which is quite common in many RL settings. BTW, as presented in the related work, using RL to optimize non-differentiable metric is not new. 2. The authors report major results on COCO in table 6. However, the performance of the baselines are far from state-of-the-art (e.g., in BLEU-4 score, usually the state of the art results are around 30%, while in table 6 the baseline result is at 20%), and the improvement from using feedback over baseline is less than 0.5% in BLEU-4, which usually is not statistically significant. 3. There is lack of detailed analysis and examples showing how feedback helps in the RL framework. e.g., given RLB(RL baseline) and RLF (w/ feedback) give similar scores, are they making similar error? Or w/feedback, the model predict very differently?

Reviewer 2



The paper presents an approach for automatically captioning images where the model also incorporates natural language feedback from humans along with ground truth captions during training. The proposed approach uses reinforcement learning to train a phrase based captioning model where the model is first trained using maximum likelihood training (supervised learning) and then further finetuned using reinforcement learning where the reward is weighted sum of BLEU scores w.r.t to the ground truth and the feedback sentences provided by humans. The reward also consists of phrase level rewards obtained by using the human feedback. The proposed model is trained and evaluated on MSCOCO image caption data. The proposed model is compared with a pure supervised learning (SL) model, a model trained using reinforcement learning (RL) without any feedback. The proposed model outperforms the pure SL model by a large margin and the RL model by a small margin. Strengths: 1. The paper is well motivated with the idea of using human in the loop for training image captioning models. 2. The baselines (SL and RL) are reasonable and the additional experiment of using 1 GT vs. 1 feedback caption is insightful and interesting. 3. The work can be great significance especially if the improvements are significantly large over the RL without any feedback baseline. Weaknesses: 1. The paper is motivated with using natural language feedback just as humans would provide while teaching a child. However, in addition to natural language feedback, the proposed feedback network also uses three additional pieces of information – which phrase is incorrect, what is the correct phrase, and what is the type of the mistake. Using these additional pieces is more than just natural language feedback. So I would like the authors to be clearer about this in introduction. 2. The improvements of the proposed model over the RL without feedback model is not so high (row3 vs. row4 in table 6), in fact a bit worse for BLEU-1. So, I would like the authors to verify if the improvements are statistically significant. 3. How much does the information about incorrect phrase / corrected phrase and the information about the type of the mistake help the feedback network? What is the performance without each of these two types of information and what is the performance with just the natural language feedback? 4. In figure 1 caption, the paper mentions that in training the feedback network, along with the natural language feedback sentence, the phrase marked as incorrect by the annotator and the corrected phrase is also used. However, from equations 1-4, it is not clear where the information about incorrect phrase and corrected phrase is used. Also L175 and L176 are not clear. What do the authors mean by “as an example”? 5. L216-217: What is the rationale behind using cross entropy for first (P – floor(t/m)) phrases? How is the performance when using reinforcement algorithm for all phrases? 6. L222: Why is the official test set of MSCOCO not used for reporting results? 7. FBN results (table 5): can authors please throw light on why the performance degrades when using the additional information about missing/wrong/redundant? 8. Table 6: can authors please clarify why the MLEC accuracy using ROUGE-L is so low? Is that a typo? 9. Can authors discuss the failure cases of the proposed (RLF) network in order to guide future research? 10. Other errors/typos: a. L190: complete -> completed b. L201, “We use either … feedback collection”: incorrect phrasing c. L218: multiply -> multiple d. L235: drop “by” Post-rebuttal comments: I agree that proper evaluation is critical. Hence I would like the authors to verify that the baseline results [33] are comparable and the proposed model is adding on top of that. So, I would like to change my rating to marginally below acceptance threshold.

Reviewer 3



Summary: The authors propose to incorporate human feedback in natural language for the task of image captioning. Specifically, they take a snapshot of a phrase-based caption model, predict captions for a subset of images, collect annotations to identify the mistakes made by the model if any, train a feedback network and ‘teach’ the model to improve over it’s earlier snapshot with this feedback using reinforcement learning. Experiments show that such feedback helps improve the performance using automatic evaluation metrics for captioning. Strengths: (a) The paper addresses a well-motivated aspect for learning - human feedback. Their results suggest that human feedback on the model’s performance is more valuable than additional caption annotations, which is intuitive and a positive outcome. (b) The authors have done a good job on collecting human feedback in a least ambiguous way to finetune the model later. Their selection of a phrase-based captioning model aids this collection procedure. (c) Experiments are comprehensive with different ablations to prove the effectiveness of the human feedback. Weaknesses: (a) Human evaluation on the quality of captions would have given a better sense of performance. Even though on a smaller scale, such evaluations throw more light than automatic correlations which have been shown to correlate poorly with respect to human judgement. Comments: (a) L69 - [3] does not use reinforcement learning. [34] seems to be doing reinforcement learning on visual dialog. (b) Table 6 -Rouge - L second row - Is the value incorrect ? Does not seem to be in same ballpark as other models. (c) Figure 2 appears incomplete. Without properly labeling (c_t, h_t, h_{t-1}, etc), it is very unclear as to how takes in the image and outputs caption phrases in the diagram. (d) Sec 3.3: The overall setup of RL finetuning using feedback has not been discussed until that point. However, this section seems to describe feedback network assuming the setup thereby raising confusion. For example, after reading L162, one might assume that feedback network is trained via reinforcement learning. L164 - ‘Newly sampled caption’ has no context. I recommend adjusting the text accordingly to setup the overall training procedure. (e) L245 - What does this sentence mean ? (f) How is decoding done at evaluation -- beam search / sampling / greedy policy ? (g) L167 - inconsistency in superscript. (h) L222 - Why such a small testing dataset, which is almost 1/20th of the training dataset ? Typos: (a) L28-29 - reinforcement learning (RL) (b) L64 - it exploits (c) L74 Policy Gradients -> policy gradients (d) L200 - typo in the equation specifying range for \lambda? References: [34] Das, Abhishek, et al. "Learning Cooperative Visual Dialog Agents with Deep Reinforcement Learning." arXiv preprint arXiv:1703.06585 (2017).